# *Arabidopsis* SEC13B Interacts with Suppressor of Frigida 4 to Repress Flowering

**DOI:** 10.3390/ijms242417248

**Published:** 2023-12-08

**Authors:** Yanqi Yang, Hao Tian, Chunxue Xu, Haitao Li, Yan Li, Haitao Zhang, Biaoming Zhang, Wenya Yuan

**Affiliations:** State Key Laboratory of Biocatalysis and Enzyme Engineering, School of Life Sciences, Hubei University, Wuhan 430062, China; yyq.yy572@foxmail.com (Y.Y.); th13872758334@163.com (H.T.); m13080664414@163.com (C.X.); lht@hubu.edu.cn (H.L.); 20210035@hubu.edu.cn (Y.L.); zht@hubu.edu.cn (H.Z.)

**Keywords:** COPII, SEC13A, SEC13B, SUF4, flowering, *Arabidopsis*

## Abstract

SECRETORY13 (SEC13) is an essential member of the coat protein complex II (COPII), which was reported to mediate vesicular-specific transport from the endoplasmic reticulum (ER) to the Golgi apparatus and plays a crucial role in early secretory pathways. In *Arabidopsis*, there are two homologous proteins of SEC13: SEC13A and SEC13B. *SUPPRESSOR OF FRIGIDA 4* (*SUF4*) encodes a C2H2-type zinc finger protein that inhibits flowering by transcriptionally activating the *FLOWERING LOCUS C* (*FLC*) through the FRIGIDA (FRI) pathway in *Arabidopsis*. However, it remains unclear whether SEC13 proteins are involved in *Arabidopsis* flowering. In this study, we first identified that the *sec13b* mutant exhibited early flowering under both long-day and short-day conditions. Quantitative real-time PCR (qRT–PCR) analysis showed that both *SEC13A* and *SEC13B* were expressed in all the checked tissues, and transient expression assays indicated that SEC13A and SEC13B were localized not only in the ER but also in the nucleus. Then, we identified that SEC13A and SEC13B could interact with SUF4 in vitro and in vivo. Interestingly, both *sec13b* and *suf4* single mutants flowered earlier than the wild type (Col-0), whereas the *sec13b suf4* double mutant flowered even earlier than all the others. In addition, the expression of flowering inhibitor *FLC* was down-regulated, and the expressions of flowering activator *FLOWERING LOCUS T* (*FT)*, *CONSTANS* (*CO),* and *SUPPRESSOR OF OVEREXPRESSION OF CO 1* (*SOC1)* were up-regulated in *sec13b*, *suf4,* and *sec13b suf4* mutants, compared with Col-0. Taken together, our results indicated that SEC13B interacted with SUF4, and they may co-regulate the same genes in flowering-regulation pathways. These results also suggested that the COPII component could function in flowering in *Arabidopsis*.

## 1. Introduction

In eukaryotes, approximately one-third of proteins are synthesized in the endoplasmic reticulum (ER) and transported to various organelles [1]. Protein trafficking is carried out via distinct vesicles between organelles. The primary transports of eukaryotic cells include retrograde transport of proteins from the Golgi to the endoplasmic reticulum (ER), anterograde transport from the ER to the Golgi, and endocytic transport from the plasma membrane to endosomes, which are mediated via coat protein complex I (COPI) vesicles, coat protein complex II (COPII) vesicles, and clathrin-coated vesicles (CCVs), respectively [2,3,4]. Previous studies have shown that the COPII complex is composed of SEC13, SEC23, SEC24, SEC31, and secretion-associated Ras-related protein (Sar1) [5,6,7,8]. Sar1 interacts with SEC23, recruiting SEC23–SEC24 heterodimers from the cytoplasm to form a “pre-budding complex” [9]. Then, heterodimers of SEC13–SEC31 bind to the outer layer of SEC23–SEC24 via an interaction between SEC31 and SEC23, resulting in the formation of COPII vesicles [10].

COPII components are highly conserved in eukaryotes. However, in higher organisms, there are multiple paralogs for each COPII component due to gene duplication. The *Arabidopsis* genome harbors two *SEC13*, seven *SEC23*, three *SEC24*, two *SEC31*, and five *Sar1* isoforms [11]. Accumulating genetic evidence has demonstrated that COPII components function in the regulation of plant growth and development. In *Arabidopsis*, the *sar1b* mutant exhibits pollen sterility, and Sar1B and Sar1C play interchangeable roles in pollen development [12,13]. *SEC23A* and *SEC23D* present typical COPII protein localization patterns and are highly expressed in the tapetum. A single mutant of *sec23a* and *sec23d* appears to have normal fertility, whereas the *sec23a sec23d* double mutant shows decreased fertility due to impaired pollen and tapetum development [14]. SEC24A may be the most critical one among the three SEC24 isoforms, and its absence causes pollen development defects, resulting in pollen transfer failure and male gametophyte lethality [15]. In *Arabidopsis*, knocking out *SEC24B* causes mild male sterility and reduced expressions of both *SEC24B* and *SEC24C* and alters male and female gametogenesis [16]. *SEC31A* and *SEC31B* are expressed in various tissues, whereas *SEC31A* is expressed at relatively low levels [17]. Both *SEC31A* and *SEC31B* regulate tapetum development in sporogenesis and are interchangeable during pollen formation [17]. Nonetheless, there have been few reports of *SEC13* in *Arabidopsis*. *SEC13A* is expressed in most regions of seedlings, and *SEC13B* exhibits specific expression [18], while their function in plant development is still unclear.

Flowering is a key to the reproductive growth stage for many plants and is controlled through several pathways [19]. In these pathways, the expression levels of *FLOWERING LOCUS T* (*FT*), *CONSTANS* (*CO*)*, SUPPRESSOR OF OVEREXPRESSION OF CO 1* (*SOC1*), and *FLOWERING LOCUS C* (*FLC*) are precisely regulated as core elements [20,21]. FT is the major component of the mobile florigen, which is transported through the phloem from the leaf to the shoot apical meristem (SAM) to promote flowering via the photoperiod pathway [22]. The photoperiod pathway outputs *CO*, whose expression is controlled via the circadian clock, to promote *FT* expression in leaf veins [23]. *SOC1* encodes a MADS-box transcription factor and integrates the photoperiod, vernalization, and autonomous pathways to activate flowering [24]. *FLC* encodes a MADS-box transcription factor, which can bind to the regulatory regions of *FT* and *SOC1* to suppress their expression and exert strong inhibitory effects on flowering [25,26]. FRIGIDA (FRI) possesses a coiled-coil domain and functions as a scaffold protein to specifically promote *FLC* expression, and the FRI complex consists of FRI, FRIGIDA LIKE1 (FRL1), FRIGIDA ESSENTIAL 1 (FES1), FLC EXPRESSOR (FLX), and SUPPRESSOR OF FRIGIDA 4 (SUF4) [27,28]. FRI recognizes and binds to the *FLC* promoter region through SUF4, subsequently recruiting the FRI complex to the *FLC* loci [28]. Nevertheless, the detailed mechanism is still unknown.

In this study, we investigated the biological function of SEC13B and compared it with SEC13A in terms of expression pattern, subcellular localization, and protein interaction. We found that a T-DNA insertion null mutant of *SEC13B* exhibited early flowering compared to the wild type (Col-0) under both long-day and short-day conditions. Additionally, we also identified that SEC13A and SEC13B could interact with SUF4 in vitro and in vivo, and the proline-rich domain was responsible for the interaction between SUF4 and SEC13 proteins. The *sec13b suf4* double mutant showed even earlier flowering, greater down-regulation of *FLC* expression, and higher up-regulation of *FT*, *CO*, and *SOC1* expression than the *sec13b* and *suf4* single mutants. Our results indicated that SEC13B could associate with SUF4 to affect flowering and provide insights into the regulatory role of the COPII vesicle in flowering in *Arabidopsis*.

## 2. Results

### 2.1. SEC13B Affects Flowering in Arabidopsis

In a mutant scan for novel genes participating in flowering regulation, a *SEC13B* gene mutant (*sec13b*, SALK_106213C) showed an obvious early flowering phenotype. The mutant originated from a T-DNA insertion in the intron of the 5′ untranslated region (UTR), which is 175 bp away from the predicted translation initiation codon (ATG) of the *SEC13B* gene (Figure 1a). In the *sec13b* mutant, the expression of *SEC13B* was completely abolished (Figure 1b). The flowering phenotype of the *sec13b* mutant was assessed primarily based on the total number of rosette and cauline leaves, using Col-0 as the control. The results showed that under long-day (LD) conditions (16 h light/8 h dark), Col-0 had approximately 13 total leaves, whereas the *sec13b* mutant displayed around 10 total leaves (Figure 1c,e). A similar result was observed under short-day (SD) conditions (8 h light/16 h dark). Compared with Col-0, the *sec13b* mutant also showed an obvious earlier flowering (Figure 1d,f). In *Arabidopsis*, there was a homolog of SEC13B, named SEC13A, which showed 89% identity and 93.7% similarity in amino acid sequence with that of SEC13B (Appendix A). It is speculated that the function of SEC31A is similar to that of SEC31B. Unfortunately, the T-DNA of the *SCE13A* mutant (*sec13a*, SALK_202621) was inserted in the 3′ UTR, and the expression of *SCE13A* was not influenced.

### 2.2. SEC13A and SEC13B Are Expressed in the Reproductive Phase, and Their Encoding Proteins Localize in the Nucleus and ER

SEC13A and SEC13B are the outer coat proteins of COPII vesicles in *Arabidopsis*. To explore the function of SEC13A and SEC13B in flowering, we analyzed the expression patterns of *SEC13A* and *SEC13B*. Total RNA from various tissues and organs was extracted, reverse-transcribed to cDNA, and analyzed using quantitative real-time PCR (qRT-PCR). The samples of seedlings and roots were collected 10 days after germination (DAG), and those of inflorescences, stems, rosette leaves, cauline leaves, and siliques were collected at 50 DAG. The results analysis revealed that *SEC13A* and *SEC13B* showed similar expression patterns, and both were expressed in all the checked tissues, though *SEC13B* was expressed higher than *SEC13A* in most tissues (Figure 2a).

In order to determine the intracellular distribution of SEC13 proteins, SEC13A-GFP and SEC13B-GFP were constructed and introduced into Col-0 protoplasts along with the ER marker, AtHDEL-mCherry, or the nuclear marker, OsGhd7-mCherry, respectively. Following transformation, the GFP signals of SEC13A and SEC13B co-localized with both AtHDEL and OsGhd7, indicating that they were localized not only in the ER but also in the nucleus (Figure 2b). These results suggest that the functions of SEC13A and SEC13B might not be restricted in the transportation from ER to Golgi.

### 2.3. SEC13A and SEC13B Interact with SUF4 In Vitro and In Vivo

To investigate the functions of SEC13A and SEC13B beyond the COPⅡ transport process in *Arabidopsis*, both proteins were employed as baits for yeast two-hybrid (Y2H) screening in an *Arabidopsis* flowering-related cDNA library. Interestingly, SUF4 (AT1G30970) was identified to interact with both SEC13A and SEC13B, and the interactions were verified in Y2H point-to-point assays (Figure 3a). Then, pull-down experiments were performed to confirm these interactions. His-tagged SUF4 (His-SUF4), GST-tagged SEC13A (GST-SEC13A), and SEC13B (GST-SEC13B) fusion proteins were expressed separately in *E. coli*. After pulling down, His-SUF4 could bind to both GST-SEC13A and GST-SEC13B instead of the negative control (GST) (Figure 3b,c).

To determine whether SEC13A and SEC13B could interact with SUF4 in plant cells, we conducted the luciferase complementary imaging (LCI) assay in *N*. *benthamiana*. SEC13A and SEC13B were fused with the N-terminal half of luciferase (nLUC), and SUF4 was fused with the C-terminal half of luciferase (cLUC). As expected, a strong fluorescence signal was found in the cLUC-SUF4/nLUC-SEC13A and cLUC-SUF4/nLUC-SEC13B co-transformations but not in the negative controls (Figure 3d,e). To further confirm these results, SEC13A, SEC13B, and SUF4 were labeled with FLAG or HA and expressed in Tobacco leaves for co-immunoprecipitation (Co-IP) assays. It was found that both SEC13A and SEC13B interacted with SUF4 in Tobacco (Figure 3f,g).

To ascertain which domain of SUF4 was required for its interaction with SEC13A and SEC13B, a series of truncated SUF4 cDNAs were constructed for testing in vitro and in vivo (Appendix A). Y2H assays demonstrated that the C-terminal fragment (amino acids 101 to 367) harboring the proline-rich domain of SUF4 (named SUF4^Pro^) was sufficient for its interaction with SEC13A and SEC13B (Appendix A). Then, strong LUC signals were detected in cLUC-SUF4^Pro^/nLUC-SEC13A and cLUC-SUF4^Pro^/nLUC-SEC13B co-transformation in Tobacco (Appendix A). Moreover, the interactions were also verified in pull-down experiments (Appendix A). All these results strongly suggest that SEC13A and SEC13B interact directly with SUF4.

### 2.4. SEC13B and SUF4 Mutations Had to Promote Flowering in Arabidopsis

Previous studies have shown that *SUF4* is involved in delaying the flowering of *Arabidopsis* [29]. To further study the role of its interaction with SEC13B in flowering regulation, we obtained a mutant of the *SUF4* gene (*suf4*, SALK_093449C). The mutant was derived from a T-DNA insertion in the fourth exon of the *SUF4* gene (Appendix A). In the *suf4* mutant, the expression of *SUF4* was completely abolished (Appendix A). Then, we generated the *sec13b suf4* double mutant by crossing. After treatment in LD conditions for 28 days, the *suf4* and *sec13b* single mutants developed 10 and 9 total leaves on average, respectively, which were less than the average of 12 in WT (Col-0). However, the *sec13b suf4* double mutant only had seven total leaves on average, which was 41% lower than that of WT (Figure 4a,b). The *sec13b suf4* mutant exhibited an earlier flowering phenotype compared to the *sec13b*, *suf4* single mutant, and Col-0 (Figure 4a,b). Similar results were observed following SD treatment for 68 days. Compared with Col-0, *suf4,* and *sec13b* mutants, the *sec13b suf4* double mutant also showed an obvious earlier flowering phenotype (Figure 4c,d). These results indicated that *SEC13B* and *SUF4* might collaborate in regulating the expression of genes associated with flowering transition in *Arabidopsis*.

### 2.5. SEC13B and SUF4 Co-Regulate Flowering-Related Genes

To characterize the mechanisms of SEC13B and SUF4 in flowering regulation, samples of Col-0, *sec13b*, *suf4,* and *sec13b suf4* were harvested at 10 DAG (16 h light/8 h dark), and the expressions of eight flowering-related genes were analyzed. The results showed that the expression of the flowering inhibitor *FLC* expression was notably decreased in sec13b, suf4, and sec13b suf4 mutants (Figure 5). Meanwhile, flowering-promoting genes, such as *FT*, *CO,* and *SOC1*, were up-regulated in these mutants (Figure 5). All these results suggested that SEC13B and SUF4 might repress flowering through the same signaling pathway in *Arabidopsis*.

## 3. Discussion

WD40 proteins, also known as WD-repeat proteins, are widely found in eukaryotes and are involved in growth, development, and many other processes [30]. These proteins have conserved motifs that are commonly 40 amino acids in length and end with Trp-Asp dimeric peptides [30]. Nowadays, several proteins containing the WD40 domain have been reported in plants. For instance, WDR5a, the homolog of human WDR5 (for WD repeat domain 5), serves as a core component of the COMPASS-like histone methyltransferase complex required for H3K4 trimethylation at the *FLC* locus and promotes *FLC* transcription level, leading to late flowering in *Arabidopsis* [31]. *Arabidopsis* COPII components SEC13A and SEC13B also contain WD40 domains and are highly homologous, with 89% identity and 93.7% similarity in amino acid sequence (Appendix A). However, their function in plant development is largely unknown. In this study, we found that besides the ER, SEC13A and SEC13B could be localized in the nucleus, and the *sec13b* mutant showed earlier flowering than Col-0. These results indicated that SEC13B might function as a regulator in the nucleus to control flowering, and our results shed the first light on how the COPII component functions in plant flowering.

SUF4 is a member of the FRI complex, which specifically recognizes and binds to the promoter region of *FLC*, recruiting the FRI complex to the vicinity of the *FLC* locus [32,33]. Subsequently, through the FRI-mediated WDR5a–ATX1 complex, H3K4me3 is introduced to the *FLC* site, promoting *FLC* expression and leading to delayed flowering in *Arabidopsis* [34,35]. In this study, we found that the SUF4 protein was localized in the ER and nucleus of *Arabidopsis* protoplasts, which is consistent with previous reports (Appendix A) [32]. We also found that SEC13A and SEC13B both interacted with SUF4, and they could also interact with each other (Appendix A). Both *suf4* and *sec13b* mutants exhibited earlier flowering than Col-0, and the double mutant *sec13b suf4* showed even earlier flowering than every single mutant. These results suggest that SEC13B and SUF4 are involved in multiple and different pathways regulating flowering, and they may also play a role in the same flowering pathway, considering their interactions in plants. By analyzing the expression of flowering-related genes, we found that *FLC* was down-regulated, and genes such as *CO*, *FT*, and *SOC1* were up-regulated in all the mutants. Based on these results, we speculate that SEC13B may facilitate the localization of SUF4 to the *FLC* locus and participate in flowering through WDR5a-mediated H3K4 trimethylation in *Arabidopsis*. Still, the essential roles of SEC13B–SUF4 interaction in flowering transition in *Arabidopsis* need to be elucidated through further analysis, such as chromatin immunoprecipitation (CHIP) assays and histone modification analysis of candidate target genes. In summary, our results suggested that SEC13B could interact with SUF4 and lead to the repression of flowering through sophisticated regulation of flowering-related genes.

## 4. Materials and Methods

### 4.1. Plant Materials and Growth Conditions

The wild type was the *Arabidopsis* ecotype Columbia (Col-0). The mutants *suf4* (SALK_093449C) and *sec13b* (SALK_106213C) were obtained from the *Arabidopsis* Biological Resource Center (ABRC), Ohio, DC, USA (http://www.arabidopsis.org/abrc/, accessed on 31 October 2023), and were then checked via PCR using the corresponding primers (Appendix A). The *Arabidopsis* plants were grown under long-day (16 h light/8 h dark) or short-day (8 h light/16 h dark) conditions at 22 °C in the greenhouse.

### 4.2. Plasmid Construction and Plant Transformation

In this study, all plasmids were assembled using the Gibson assembly method [36]. The PCR products amplified with primer pairs and the target vectors digested via appropriate restriction endonucleases were purified and quantified using NanoDrop 2000 (Thermo Fisher Scientific, Waltham, MA, USA). DH5α was used for construction, and DE3/BL21 was used for protein expression. The transient expression constructs were transformed into *N*. *benthamiana* using *Agrobacterium tumefaciens* strain GV1301 (containing the pSoup-p19 plasmid).

### 4.3. qRT-PCR Assays

*Arabidopsis* plants were cultivated in a controlled environmental chamber at a temperature of 22 °C, following long-day conditions (16 h light/8 h dark) for 10 days. Subsequently, whole seedlings were collected to facilitate the extraction of RNA. First-strand cDNA synthesis was carried out using the Reverse Transcription Kit (Invitrogen, Carlsbad, CA, USA). Then, three technical replicates for each RNA sample were used in quantitative RT-PCR analysis, which was carried out in the Bio-Rad CFX384 Touch real-time PCR detection system with *TUB2* as an internal reference. The relative mRNA expression levels were determined using the 2−∆∆CT method. Each sample was analyzed three times, and only one replicate was present. All primers used for expression analysis are listed in Appendix A.

### 4.4. Yeast Two-Hybrid Assays

The fragments containing different domains of *SUF4*, *SEC13A, and SEC13B* were amplified from cDNAs of Col-0 using gene-specific primers (Appendix A). Then, the products were inserted into pGBKT7 and pGADT7-Rec (Clontech, Shiga, Japan) vectors via restriction endonuclease digestion and ligation. The paired constructs were co-transformed into the yeast strain AH109. Co-transformants were plated on a selective medium lacking Trp, Leu, His, and Ade (–WLHA) or a non-selective medium lacking Trp and Leu (–WL) for 3 days at 30 °C. Empty vectors were used as negative controls. Each yeast two-hybrid assay was repeated at least twice with similar results, and only one replicate was presented.

### 4.5. Pull-Down Assays

The SEC13A, SEC13B, and SUF4 CDS were obtained from the Col-0 cDNA using gene-specific primers (Appendix A). The fragments were constructed into the pGEX4T-1 or pET28a expression vector [37]. To detect the SEC13A–SUF4 and SEC13B–SUF4 interactions in vitro using pull-down assays, roughly equal amounts of GST-SEC13A, GST-SEC13B, and His-SUF4 were incubated at 16 °C for 14 h. The protein pull-down assays were conducted in accordance with the manufacturer’s instructions, utilizing glutathione ligand resin affinity purification from Sigma-Aldrich (Darmstadt, Germany) [38]. In brief, after induction with isopropyl-b-D-thiogalactopyranoside, *E. coli* (DE3/BL21) cells that carried GST-SEC13A, GST-SEC13B, His-SUF4, or GST were harvested via centrifugation, then resuspended in 1.0 mL binding buffer (20 mM Tris-HCl, pH 7.5, 150 mM NaCl, 0.1% Triton X-100, 10% glycerol, 1 mM phenylmethylsulphonyl fluoride, and 1× protease inhibitor cocktail) and subjected to sonication. The proteins were detected using Western blot analysis using an anti-GST antibody (PM013, MBL Beijing Biotech Co., Ltd., Beijing, China) or an anti-His antibody (M136-3, MBL Beijing Biotech Co., Ltd., Beijing, China).

### 4.6. LCI Assays

The LCI assays were performed following the established protocol [39]. The full-length fragments of SEC13A and SEC13B cDNA were in-frame fused with the N-terminal half of luciferase (nLUC), whereas the full-length and different domain-containing fragments of SUF4 cDNA were in-frame fused with the C-terminal half of luciferase (cLUC). Subsequently, the constructed vectors were introduced into competent *Agrobacterium tumefaciens* GV3101 (containing the pSoup-p19 plasmid). *Agrobacterium* cells containing various expression vectors were co-infiltrated into *N*. *benthamiana*. The plants were kept in darkness at 22 °C for 24 h, and the interaction signals were monitored at 48 h after infiltration. During the examination, 1× D-fluorescein (Biotium, Fremont, CA, USA) was consistently applied to the entire leaf. The fluorescent images were captured with a low-light, cooled charge-coupled device (CCD) imaging system (Tanon 5200, Beijing, China).

### 4.7. Co-Immunoprecipitation

The fragments of SEC13A, SEC13B, and SUF4 were amplified using gene-specific primers and ligated into the pMDC32-3× FLAG or the pMDC32-3× HA vector (Appendix A) [40]. Then, the constructs were introduced into *Agrobacterium tumefaciens* strain GV3101. *Agrobacterium*-mediated transformation was performed via infiltration into *N*. *benthamiana* using needleless syringes. Proteins were extracted using an extraction buffer (50 mM Tris-HCl, pH 7.5, 150 mM NaCl, 1 mM EDTA, 1 mM DTT, 10% glycerol, 1 mM PMSF, and 0.1% Triton X-100). The proteins were incubated with anti-HA beads (L-1009, BIOLINKEDIN, Shanghai, China) and then washed as previously described [41]. Western blot analyses were performed using anti-Flag (M185-3S, MBL Beijing Biotech Co., Ltd., Beijing, China) and anti-HA antibodies (M132-3, MBL Beijing Biotech Co., Ltd., Beijing, China). The co-immunoprecipitation analysis was biologically repeated twice with similar results, and only one replicate was presented.

### 4.8. Subcellular Localization Assays

Transient expression assays were performed in *Arabidopsis* protoplasts to determine the subcellular localization of *SEC13A*, *SEC13B,* and *SUF4*. The complete cDNA sequences of SEC13A, SEC13B, and SUF4 were cloned into the transient expression plasmid pAN580 to produce the *p35S::*SEC13A-GFP, *p35S::*SEC13B-GFP, and *p35S::*SUF4-GFP fusion constructs. The *Arabidopsis thaliana* protoplast isolation and transient expression evaluation were performed according to established procedures [42]. Fluorescent signals were visualized utilizing the ZEISS LSM980 laser scanning confocal microscope.

### 4.9. AGI Locus Numbers

Sequence data from this article can be found in the *Arabidopsis* Genome Initiative or GenBank/EMBL databases under the following accession numbers: *SEC13A* (*At2g30050*), *SEC13B* (*At3g01340*), and *SUF4* (*At1g30970*).

## 5. Conclusions

*SEC13B* in *Arabidopsis*, homologous to yeast *SEC13*, encodes a protein presumed to be a constituent of COPII vesicles involved in vesicle-specific transport from the ER to the Golgi apparatus. The *Arabidopsis* transcription factor *SUF4* encodes a C2H2-type zinc finger protein capable of recognizing and binding to the promoter region of *FLC*, promoting *FLC* expression, and leading to delayed flowering. In this study, we found that *SEC13B* was involved in flowering through mutant scanning. We also found and verified that SEC13B could interact with SUF4, a member of the FRI complex, in flowering regulation. Both *sec13b* and *suf4* single mutants flowered earlier than Col-0, whereas the *sec13b suf4* double mutant flowered even earlier than all the others. Several flowering-related genes were regulated in a similar pattern in *sec13b*, *suf4* single, and *sec13b suf4* double mutants. These results suggested that SEC13B might interact with SUF4 and regulate the same genes through the same pathway(s) in flowering; meanwhile, they were also probably involved in different pathways considering the severe phenotype of the *sec13b suf4* double mutants. The results here shed new light on the regulatory mechanisms governing flowering transitions mediated with SEC13B and SUF4.

## Figures and Tables

**Figure 1 ijms-24-17248-f001:**
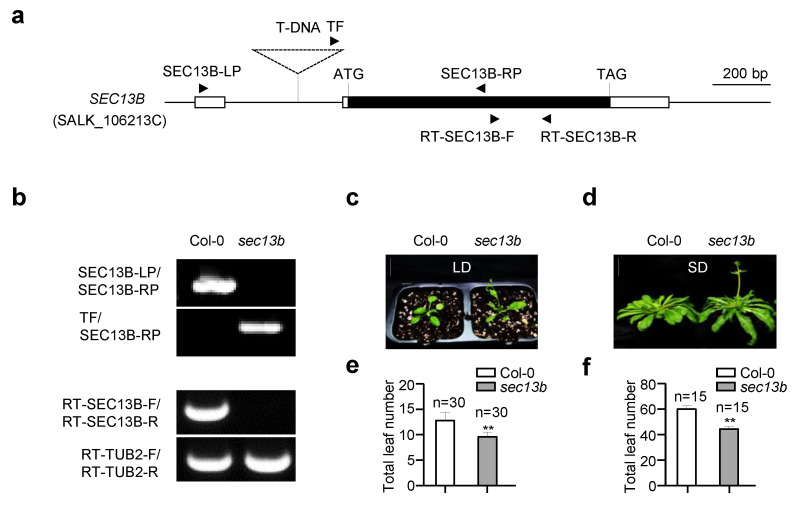
*SEC13B* affects flowering in *Arabidopsis*. (**a**) Gene structure of *SEC13B*. The white boxes indicate UTRs, the black box indicates exons, the line between the boxes indicates introns, and the dashed triangle indicates the inserted T-DNA. Black arrows indicate the positions and directions of the primers. SEC13B-LP, SEC13B-RP, and TF were used for the identification of the *sec13b* mutant (SALK_106213C); RT-SEC13B-F/R and RT-TUB2-F/R were used for reverse transcription PCR (RT-PCR). (**b**) Genotyping and transcription analysis of the Col-0 and *sec13b* mutants via PCR and RT-PCR, respectively. *TUBULIN 2* is used as a control. (**c**,**e**) The phenotype of Col-0 and *sec13b* at the flowering stage in LD (long day) and total leaf number analysis of Col-0 and *sec13b* at the flowering stage in LD. Bar = 2 cm. The symbol “n” indicates the number of investigated plants. A significant difference was detected between Col-0 and mutant at *p* < 0.01 (indicated with “****”). (**d**,**f**) Phenotype of Col-0 and *sec13b* at the flowering stage in SD (short day) and total leaf number analysis of Col-0 and *sec13b* at the flowering stage in SD. Bar = 2 cm. The symbols “n” indicate the numbers of investigated plants. A significant difference was detected between Col-0 and mutant at *p* < 0.01 (indicated with “****”).

**Figure 2 ijms-24-17248-f002:**
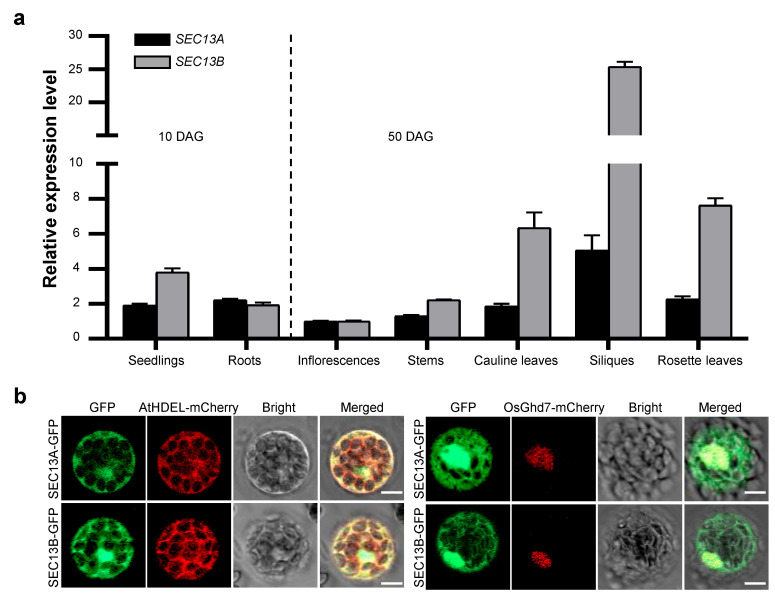
Expression pattern of *SEC13* genes and subcellular localization of SEC13 proteins. (**a**) Expression pattern of *SEC13A* and *SEC13B* in *Arabidopsis*. Seedlings and roots were collected 10 days after germination (DAG); inflorescences, stems, rosette leaves, cauline leaves, and siliques were collected at 50 DAG. Values are presented as means ± SD (n = 3). (**b**) Subcellular localization of SEC13A and SEC13B. AtHDEL-mCherry was used as an ER marker. OsGhd7-mCherry was used as a nuclear marker. The images were taken in green (GFP fluorescence), red (mCherry fluorescence), and bright and red–green combination (merged) channels, respectively. Bar = 20 µm.

**Figure 3 ijms-24-17248-f003:**
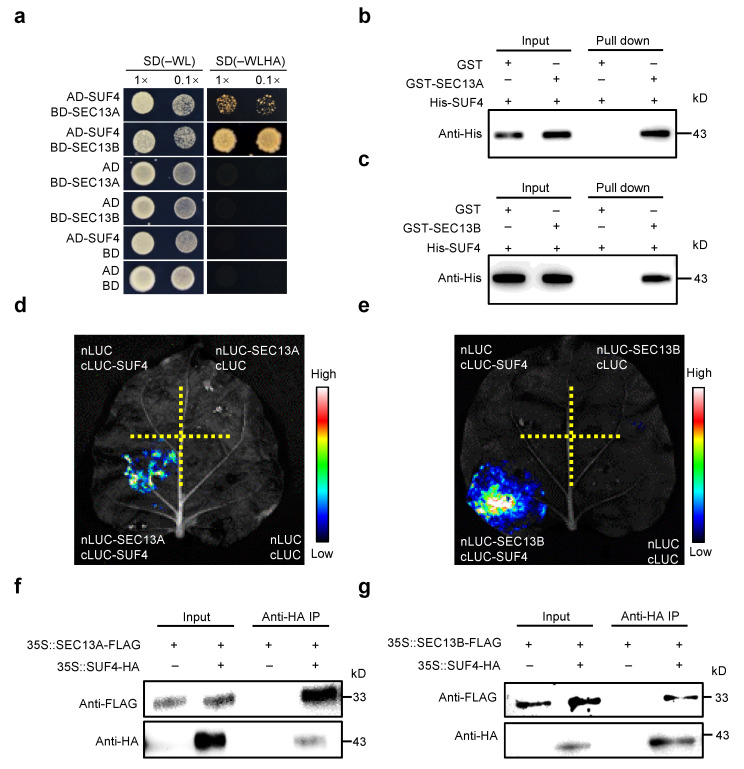
SEC13A and SEC13B directly interact with SUF4. (**a**) SEC13A and SEC13B interact with SUF4 in yeast. The transformed yeast cells were spotted on a stringent selective medium lacking Trp, Leu, His, and Ade (–WLHA) or a non-selective medium lacking Trp and Leu (–WL; control). (**b**,**c**) Expressed SEC13A and SEC13B interact with SUF4 in vitro. All protein samples were immunoprecipitated with anti-GST beads and immunoblotted with anti-His antibodies. (**d**,**e**) LCI assays show that SEC13A and SEC13B interact with SUF4 in *N. benthamiana*. (**f**,**g**) Co-IP assays validate that SEC13A and SEC13B interact with SUF4 in *N*. *benthamiana*. The symbols “+” and “−” represent the presence and absence of the corresponding proteins.

**Figure 4 ijms-24-17248-f004:**
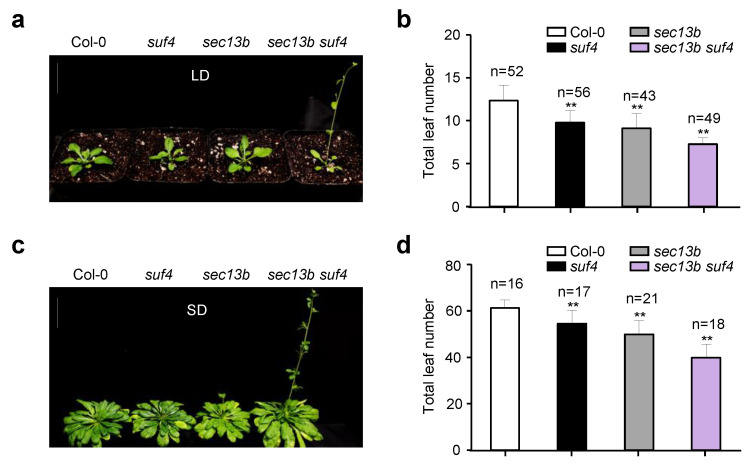
Mutants of *SEC13B* and *SUF4* promote flowering in *Arabidopsis*. (**a**,**b**) Phenotypes of Col-0, *suf4*, *sec13b*, and *sec13b suf4* at the flowering stage in LD. Bar = 2 cm. (**c**,**d**) Phenotypes of Col-0, *suf4*, *sec13b*, and *sec13b suf4* at the flowering stage in SD. Bar = 2 cm. The symbol “n” indicates the number of plants. Significant differences were detected between Col-0 and mutants at *p* < 0.01 (indicated with “****”).

**Figure 5 ijms-24-17248-f005:**
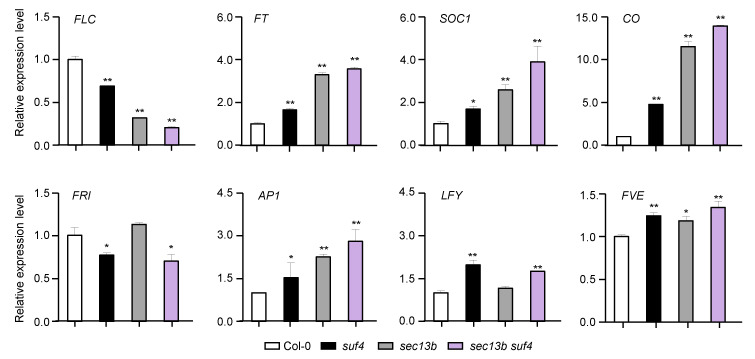
SEC13B and SUF4 co-regulate flowering-related genes in *Arabidopsis*. Relative transcript levels of flowering-related genes were determined via qRT-PCR analysis. Values are presented as means (±SD) of three replicates. Significant differences were detected between Col and mutants at 0.01 < *p* < 0.05 (indicated with “***”) and *p* < 0.01 (indicated with “****”).

## Data Availability

Data is contained within the article and supplementary material.

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
