# Peer review of "Arabidopsis SEC13B Interacts with Suppressor of Frigida 4 to Repress Flowering"

_ijms, 2023, doi:10.3390/ijms242417248_

Round 1
Reviewer 1 Report
Comments and Suggestions for Authors
The title of the study seems interesting, the study is focused on the Arabidopsis SEC13B Interact with SUF4 to Repress Flowering”. The authors identified that sec13b mutant exhibits early flowering, they also identified that SEC13A 21and SEC13B could interact with SUF4 in vitro and in vivo. No doubt, the study is aligned with the scope of the journal and provides some interesting insights into the literature. While there are interesting aspects to the results, I also believe there are some essential weaknesses. Therefore, I have several reservations about this paper and which are summarized as follows:.
Main Points:
It is not clear to explain the contribution of this paper. In other word, it is by no means clear why this is an interesting area of study and the paper needs a much stronger motivation. Because there have been considerable studies on the Arabidopsis SEC13B Interact with SUF4 to Repress Flowering. Moreover, all methods used in this study are the existing approaches with no contributed adjustments by authors. Try to answer questions like, who would use your results? Why your results and not others in the literature? I did not find any novel stories from the paper.
I would like to see more discussions of the literature so that I can clearly identify the article relating to the author's ideas and study.
I do not see detailed or interpretation of the author's methodologies. The methods are existing methods. In "Methodology", I suggest authors provide explanations on what and why the methodology was applied in this study.
There are too many simple conclusive statements in the abstract, which are well-known to people who are involved in this area. You should first lay out a background information description.
The conclusion part of the study does not support the results provided. Moreover, what would be the potential benefits of this study, who used these results….? Is still a question.
Minor Points:
I found various grammatical mistakes in the manuscript.
It is recommended that a native speaker must review the manuscript before publication. Moreover, the author may forget to cite a recent study on a similar topic.
Comments on the Quality of English LanguageMinor improvement required
Author Response
Dear reviewer.
Thank you very much for taking the time to review this manuscript. Please find the detailed responses below attachment.

Reviewer 2 Report
Comments and Suggestions for Authors
The authors of the manuscript found through knockout mutants that SEC13b protein, a protein involved in vesicle-specific transport from the ER to Golgi apparatus, is involved in flowering, as a knockout mutant, not expressing SAC13b was flowering faster than wild type plant. Next, they found that SEC13b interacts with SUF4 protein, a zinc finger protein promoting FLC expression, thus delaying the flowering. SUF4 knockout mutant also flowered faster than the wild type, and double knockout of both SAC13b and SUF4 was flowering even sooner than single mutants. The authors also analyzed the relative expression level of genes involved in delaying and inducing flowering in wild-type plants, single mutants, and double mutants. They concluded that SEC13B interaction with SUF4 regulates the expression of these genes.
The methods chosen by the authors were proper, and the results look solid and strongly support the conclusion. The discussion provides a plausible explanation of the observed phenomena. The scientific significance of the manuscript is, in my opinion, very high, and I recommend publishing this manuscript.
Minor comment
The only thing I am missing in this paper is a graphical model of the interactions between investigated proteins leading to sooner or delayed flowering. Even when some stages are unclear, such a model would summarise the discovery well and indicate the missing points for further investigations.
Author Response

(The authors gave the same response as above.)

Round 2
Reviewer 1 Report
Comments and Suggestions for Authors
The authors put energies in revision of the paper. The revised paper has significant improvement in the quality.